# Identifying indicators influencing emergency department performance during a medical surge: A consensus-based modified fuzzy Delphi approach

Egbe-Etu Etu [1☯¤]*, Leslie Monplaisir[1‡], Celestine Aguwa[1☯], Suzan Arslanturk[2‡], Sara Masoud[1‡], Ihor Markevych[3], Joseph Miller[4☯]

1 Department of Industrial & Systems Engineering, Wayne State University, Detroit, Michigan, United States of America, 2 Department of Computer Science, Wayne State University, Detroit, Michigan, United States of America, 3 School of Computer Science, Carnegie Mellon University, Pittsburgh, Pennsylvania, United States of America, 4 Departments of Emergency Medicine and Internal Medicine, Henry Ford Hospital, Detroit, Michigan, United States of America

☯ These authors contributed equally to this work.
¤ Current address: Department of Marketing & Business Analytics, San Jose State University, San Jose, California, United States of America
‡ LM, SA and SM also contributed equally to this work.
* egbe-etu.etu@sjsu.edu

**Data Availability Statement:** All relevant data are within the paper and its Supporting Information files.

## Abstract

During a medical surge, resource scarcity and other factors influence the performance of the healthcare systems. To enhance their performance, hospitals need to identify the critical indicators that affect their operations for better decision-making. This study aims to model a pertinent set of indicators for improving emergency departments' (ED) performance during a medical surge. The framework comprises a three-stage process to survey, evaluate, and rank such indicators in a systematic approach. The first stage consists of a survey based on the literature and interviews to extract quality indicators that impact the EDs' performance. The second stage consists of forming a panel of medical professionals to complete the survey questionnaire and applying our proposed consensus-based modified fuzzy Delphi method, which integrates text mining to address the fuzziness and obtain the sentiment scores in expert responses. The final stage ranks the indicators based on their stability and convergence. Here, twenty-nine potential indicators are extracted in the first stage, categorized into five healthcare performance factors, are reduced to twenty consentaneous indicators monitoring ED's efficacy. The Mann-Whitney test confirmed the stability of the group opinions ($p < 0.05$). The agreement percentage indicates that ED beds (77.8%), nurse staffing per patient seen (77.3%), and length of stay (75.0%) are among the most significant indicators affecting the ED's performance when responding to a surge. This research proposes a framework that helps hospital administrators determine essential indicators to monitor, manage, and improve the performance of EDs systematically during a surge event.

**Funding:** Funding: This work was supported by Blue Cross Blue Shield of Michigan Foundation (Grant #: 002934.PIRAP, 2020). Funder website: https://www.bcbsm.com/foundation/index.html Initials of the authors who received the award: JM, EEE, CA, LM The funding agency had no role in the study design, analysis, or decision to publish.

**Competing interests:** The authors have declared that no competing interests exist.

# Introduction

In times of mass casualty or public health emergency, health care facilities are likely to face a massive influx of patients [1]. During such periods, hospitals play essential roles in delivering health care services to patients. As hospitals are not only directly subjected to the consequences of the mentioned catastrophic events, they are also required to sustain and even increase their capacity to meet the increased disaster-originated demands [2, 3]. The concept of surge capability in healthcare is well-defined in various literature [4–7]. Medical surge capability refers to health care systems' ability to evaluate and provide care for a significantly increased volume of patients—one that challenges or exceeds the specified standard operating capacity of the system. The requirements for a surge may extend beyond direct patient care, including tasks such as extensive laboratory studies or epidemiological investigations of viruses and vaccines [8]. The emergency department (ED) is the front line of the health system for many patients, and it is the hospital's first department that exceeds capacity during public health emergencies.

Globally, the COVID-19 pandemic has exposed the flaws of our health care system, leading to disruptions. Hospital facilities have significantly been overwhelmed by the surge in patients requiring medical attention. For instance, reports in Italy indicate that the intensive care units were at a point of collapse with patients' influx [9]. Equipment and other resources are overused or unavailable due to the increasing rate of patients seeking care. All these disruptions impact the performance of EDs and ultimately influence the community's health outcomes in the face of disaster. Performance management is vital for improving the quality of service provided to patients in the hospital. For most patients, EDs are the first point of entry into the hospital, and a compromise in the quality of service will negatively affect the incoming, existing ED, and hospitalized patients, resulting in an increase in wait-times, patient boarding, morbidity, and mortality rates [10–12]. A diverse variety of quality indicators control the performance of different hospital departments, such as EDs. [13] postulate that *"a quality indicator is a measure relating to aspects of the healthcare system, such as the resources required to provide care, how care is delivered or the outcomes of care."* Indicators may be categorized both by the domains of quality encompassed and their relationship to the healthcare system's structure, processes, or outcomes. Concurrently, there is growing interest internationally towards developing and refining quality indicators in the ED [14–16].

Administrators used these quality indicators to evaluate the performance of the EDs during regular operations and identify areas needing improvement. Thus, the decision is usually taken based on each indicator's process factors and what they measure. Selecting the most appropriate healthcare quality indicator is essential, as poor indicator selection and application may result in unintended consequences [13]. It is evident from past studies that multiple quality indicators are tracked in the ED to provide administrators an idea of the ED's performance during normal operating conditions [17–21]. In a surge event, the nature of ED's performance is different as administrators need to address the increased logistical and operational constraints while managing their usual responsibilities, such as treatment of incoming patients. Furthermore, patients' treatment processes and flow may be much more complicated and require more resources, leading to additional strain on ED staff. There is a need to identify a list of indicators that can be tracked and monitored to help administrators efficiently monitor and evaluate the ED's performance while responding to a medical surge.

Indicator identification is complex since such indicators depend on different healthcare performance factors associated with internal and external ED operational processes. The complexity of the problem requires stakeholders' opinions (i.e., medical professionals) on the subject as they are often involved in the decision-making process [22]. Due to the problem complexity and the large number of indicators to consider, especially in healthcare, the

indicator selection process may suffer from vagueness and human mistake caused by doubts, ignorance, and inconsistencies as seen in the literature [5, 6, 19, 23]. Therefore, there is a need for a robust analytical and data-driven decision support tool to address the inaccuracy within human reasoning while quantifying linguistic variables.

In the context of healthcare research, consensus-based techniques are widely used because of their presumed capacity to extract collective knowledge from medical professionals, thereby enabling better decision-making within grey areas [24]. S1 Table in S1 File presents the pros and cons of different consensus-based methods. This paper centers on proposing a robust consensus-based technique (i.e., modified fuzzy Delphi method) for identifying the most critical indicators to the administrators that "monitor the ED performance." Also, the identified critical indicators will help facilitate the decision-making process in the ED while responding to a surge event.

## Methods

We developed a modified fuzzy Delphi method (Fig 1) for identifying and selecting the most critical indicators needed to monitor the ED's performance during a medical surge event.

### Stage 1: Study organization and questionnaire preparation

Before the literature review, multiple visits to Henry Ford ED were conducted to enable the researchers to assess the current state of the practice. These visits empowered us to interview

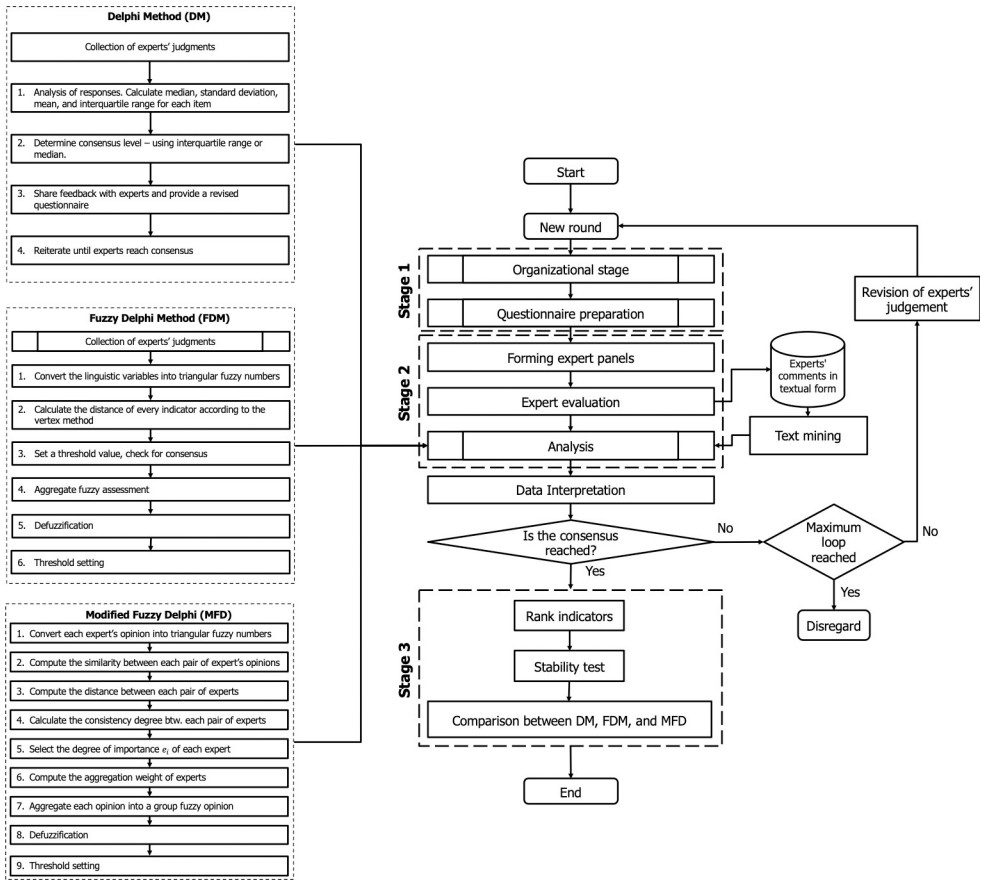

**Fig 1. The flowchart of the modified fuzzy Delphi (MOFD) method.**

medical professionals with managerial experience and hear their thoughts on critical indicators that influence the EDs' performance during a surge.

**Potential list of indicators.** A systematic review of the scientific literature is done to extract ED indicators. We explored three primary databases, namely Scopus, Google Scholar, and PubMed, using keywords such as healthcare quality metrics, emergency surge conditions, emergency department performance indicators, emergency department efficiency, and hospital key performance indicators. The identified studies contained experimental investigations about measuring the efficiency of hospitals, emergency departments, and other units within the hospital. The extracted indicators are categorized into healthcare performance factors. The healthcare performance factors are standardized, evidence-based measures of health care quality that can be used with readily available hospital data to measure and track performance and outcomes.

**Questionnaire development.** A three-section questionnaire survey is designed to identify the relevant set of indicators. While the first section covers the demographic information and the specialty of the respondents, the second section assesses the ED's performance indicators with a 5-point Likert scale (i.e., 1 = strongly disagree to 5 = strongly agree), and the third section covers respondent's comment(s) concerning ED performance during a surge. The respondents could also write their arguments for or against the proposed indicators, supporting their opinion with relevant references from the literature. One round of pretesting is performed to ensure the quality and comprehension of the questionnaire after development. The goal of pretesting is to increase the survey's reliability and validity and minimize potential errors while improving data quality. The pretesters consist of stakeholders who have worked in Michigan's healthcare agency and do not constitute the sample expert population of the study.

## Stage 2: Expert panel formation, evaluation, and analysis

The expert panel of medical professionals is formed and clustered into small subgroups based on their specialties in this stage. Each medical expert independently completes the questionnaire as they are anonymous. In this study, the inclusion criteria for the experts are healthcare-related specialization, familiarity with emergency medicine, and a minimum of three years of experience. The expert panel's role is to review the potential indicators, provide comments, and rate each indicator with regards to its usefulness in assessing the ED's performance during a surge event. All the potential indicators are formatted into a questionnaire to be completed and returned electronically. Individualized Qualtrics links are used to electronically distribute the surveys to respondents in a double-blind format. The double-blind format eliminates any bias resulting from the panel members knowing each other and the researchers.

**Analysis.** Three consensus-based methods for analysis were implemented in this study: the Delphi, Fuzzy Delphi, and Modified Fuzzy Delphi methods. The Delphi method predicts future scenarios or events through expert assessment [25]. Although DM has been widely used in the healthcare systems [19, 26–28], it suffers from some problems, mainly in counting the fuzziness of expert opinions into account. The fuzzy Delphi method (FDM) was developed to solve the challenges experienced with DM and provide exact numerical values for the comparison ratios when evaluating a given subject. The FDM combines the DM and fuzzy set theory to incorporate the vagueness and uncertainty of expert responses into the modeling process, thereby addressing the inaccuracy in human reasoning.

Although FDM was a significant improvement in consensus-based decision making, it suffers from the following. First, there is a distortion of experts' opinions when represented by fuzzy numbers causing difficulties in decision-making. Second, loss of information occurs when only distance measure (such as Euclidean distance) is used to calculate the weight of an

expert's opinion with other opinions—lastly, the lack of reasoning behind expert scores [29]. Hence, our study proposes the modified Fuzzy Delphi (MOFD) method to address these issues by combining the similarity and distance measures, which are essential indices to achieve consistency among aggregated consensuses.

The MOFD approach presented in Fig 1 is summarized as follows: suppose that $O = \{O_n | n = 1, \bar{N}\}$ is a set of $N$ healthcare performance factors, where each factor represents a finite set of indicators (i.e., $P_n$), $I_n = \{I_{np} | p = 1, \bar{P}_n\} \forall O_n \in O$. We invite a group of healthcare experts, $E$ of size $k$, to analyze each indicator, $E_{np} = \{E_{npk} | k = 1, \bar{K}_{np}\} \forall I_{np} \in I_n$. Based on their experience and knowledge, each expert uses the Likert five-level scale to evaluate the given questionnaire survey. Since the experts remain anonymous (i.e., there is no physical information exchange among them), a random distribution of the responses for each indicator is the most probable. Their responses (i.e., quantitative data) are converted from a Likert five-level scale to a set of fuzzy numbers and then aggregated to obtain a group opinion. A level of agreement (i.e., a threshold) for each indicator is obtained. Any indicator that exceeds the threshold is accepted, while those below the threshold are revised and a new round conducted. A text mining method is used to analyze expert comments (qualitative data) and confirm why certain items are rated high or low (see the S1 File for a detailed explanation of DM, FDM, and MOFD methods).

## Stage 3: Determine response stability, ranking the indicators, and comparison analysis

In this stage, first, we applied the Mann-Whitney $U$ test to determine whether a difference between the data of two fuzzy Delphi rounds has statistical significance, thereby testing for stability of the data ($p < 0.05$). According to [30], the $U$ test works with paired data of the same group of individuals as in a "before and after" setting, making it suitable for our study. Second, each indicator is ranked using a permutation operation $X \in R^n$, where $X$ is a permutation that defines the ranking operation. Lastly, we compared the results of the three consensus-based methods.

The study was approved and granted an exemption from the Wayne State University institutional review board (IRB #: IRB-19-11-1418-B3 Expedited/Exempt Review-EXEMPT) because we used de-identified data to investigate the indicators. Informed consent (i.e., written) was obtained from all the medical professionals who participated in the questionnaire survey.

## Results

### Potential list of indicators

Following an extensive literature review, site visits (examining the hospital's database), and interviews with experts, 29 potential indicators were identified. Using the quality framework provided by the Agency for Healthcare Research and Quality [31] and the key performance indicator taxonomy described by [32], the identified potential indicators were categorized into the following healthcare performance factors: Capacity, Temporal, Quality, Outcomes, and Financial Expenditures. The healthcare performance factors enabled us to capture the hospital's ability to provide emergency care or services to all severity levels of patients. Table 1 summarizes the list of potential indicators.

As displayed in Table 1, the first quality factor capacity covered demand or supply of care, which is highly prioritized during a disaster or pandemic. Additionally, it described the ability of the ED to meet the patient's demands with the available resources. The second quality

**Table 1. List of potential indicators.**

| Healthcare performance factors | Indicators |
|---|---|
| Capacity | ED beds |
| | ICU beds |
| | Physician staffing |
| | Midlevel provider staffing |
| | Nurse staffing |
| | Patient acuity level |
| | Physician staffing per patient seen |
| | Nurse staffing per patient seen |
| | Backup physician |
| | Backup nurse |
| | Patient care compromised |
| | Medical support personnel |
| Temporal | High acuity |
| | Low acuity |
| | Admit ED LOS < 6 hours |
| | Discharge ED LOS < 4 hours |
| | Time to triage |
| | Time to start of treatment |
| | Time to ED bed |
| | Time to treatment condition |
| Quality | Employee fatigue |
| | Employee satisfaction |
| | Medical errors |
| Outcomes | Patients hospitalized |
| | Patient transfers |
| Financial expenditures | Increase diagnostic test |
| | Increase ED treatment |
| | Increase ED revenue |
| | Increase in non-labor cost |

Note: **ED,** Emergency Department; **ICU,** Intensive care unit; **LOS,** Length of stay.

factor, temporal, is based on the waiting or processing time patients receive care. Medical surges cause a dramatic increase in the time that patients wait to see a nurse or doctor, which leads to high morbidity due to delayed diagnoses and treatment. Furthermore, a higher number of patients leave without being seen, which affects their health outcomes and could lead to the spread of infection within the community.

The third factor is quality, centered on staff satisfaction and medical errors in the ED when delivering care to patients. With a medical surge, the influx of patients can lead to fatigue and higher medical errors by frontline clinical staff. The fourth factor is outcomes, which during a surge, showed increased rates of hospitalization, increased length of stay in the hospital, and strain on the healthcare system. Finally, the financial factor encompassed the operating cost and expenses incurred to treat patients during a medical surge, which changes significantly from usual financial forecasting. These identified indicators are used to develop the questionnaire survey. A pretest panel of 17 respondents consisting of academicians, physicians, registered nurses, and a medical director is used to validate, check for errors, and ambiguity in the questionnaire.

## Findings of the analysis

Two rounds of panel surveys were conducted. In the first and second rounds, a total of 45 (100%) and 23 (51.11%) responses out of 45 possible professionals were received, respectively. In the second round, we observed a lower response rate from the respondents due to the increasing activity of patients seeking care in the ED during the 2019–2020 flu season. In each round, the researchers replaced the missing values with "Neutral" to avoid errors in the analysis. The demographic characteristics of the Delphi members are displayed in S2 Fig and S4 Table in S1 File.

The majority of the medical professionals that constitute the panel (37, or 82.22%) work in the ED. This includes 14 physicians (31.11%), 6 resident physicians (13.33%), 10 registered nurses (22.22%), 2 midlevel providers (4.44%), 1 medical director (2.22%), 3 emergency system administrator (6.67%), and 1 paramedic (2.22%). The other members of the panel (8, or 17.78%) work in the hospital and have vast experience in ED operations management during a surge. The respondents have a median of 7 years' experience in ED processes and operations during a surge, with a minimum of 3 years and a maximum of 11 years' experience. The expert panel had a total of 304 years of ED-related experience (S5 Table in S1 File), and the value is used to obtain a weighting for each expert (degree of importance), which we used for aggregating the expert opinion. The questionnaire survey is sent out to the medical professionals to assess the importance of each indicator.

As illustrated in S6 Table in S1 File, the first-round results of the DM process revealed a lack of agreement on the five suggested healthcare performance factors. 20 out of 29 indicators did not achieve consensus amongst the medical professionals. Furthermore, we discovered that the Delphi method does not adequately quantify the linguistic terms in the questionnaires and cater to the ambiguity in expert responses. This finding will cause the researchers to conduct multiple rounds, often tedious and time-consuming, especially when dealing with hospital staff.

The FDM and MOFD first-round results are presented in Table 2, which in contrast, showed a higher level of agreement than the DM model. A total of 16 and 12 indicators achieved consensus in the first-round FDM and MOFD process, respectively. The text-mining results of the qualitative data presented in Table 3 revealed that the average negative intensity score of the reviews on capacity is 2.1%. Respondent's concerns were on available ED space to treat or triage patients as some hospitals have small spaces allocated to the ED. Temporal-related factors had an average negative sentiment score of 2.7%, with comments on poor communication of waiting time to patients, which often leads to misunderstandings, patients leaving without being seen, and patient dissatisfaction adversely affecting the ED's performance. Quality-related factors had a negative score of 1%. Most of the comments focused on providing quality care when patients outnumber the available ED resources as staff gets overwhelmed to avoid medical errors. Outcome and financial factors had low scores, with respondents' comments indicating that EDs rarely transfer patients and experience increased testing during a surge.

The results of the first-round analysis, mode of consensus, and feedback sorted by each indicator are sent to the medical professionals. The indicators that did not achieve consensus amongst the panel members are used to create a revised survey questionnaire for the second round for final decision-making and agreement. After completing the second-round survey and analysis, five indicators in the FDM process and eight indicators in the MOFD process out of the 17 indicators achieved consensus and are recognized as suitable indicators for monitoring the ED's performance when responding to a medical surge (see Table 4).

The consensus-based process is considered complete when agreement and stability levels have been attained, as conducting another round would not significantly change the results.

**Table 2. First-round results for FDM & MOFD methods.**

| Healthcare performance factors | Metrics | FDM | | MOFD | |
|---|---|---|---|---|---|
| | | Avg. of fuzzy numbers | Consensus (threshold >64) | Avg. of fuzzy numbers | Consensus (threshold >53) |
| Capacity | ED beds | 0.128 | 95.556* | 0.113 | 77.879* |
| | ICU beds | 0.152 | 91.111* | 0.167 | 75.063* |
| | Physician staffing | 0.33 | 35.556 | 0.307 | 49.958 |
| | Midlevel provider staffing | 0.348 | 46.667 | 0.328 | 50.148 |
| | Nurse staffing | 0.218 | 86.667* | 0.167 | 62.184* |
| | Patient acuity level | 0.546 | 51.111 | 0.586 | 47.281 |
| | Physician staffing per patient seen | 0.594 | 44.444 | 0.647 | 47.618 |
| | Nurse staffing per patient seen | 0.647 | 80.0* | 0.709 | 52.943 |
| | Backup physician | 0.486 | 62.222 | 0.464 | 48.229 |
| | Backup nurse | 0.549 | 57.778 | 0.56 | 45.711 |
| | Patient care compromised | 0.628 | 80.0* | 0.668 | 51.371 |
| | Medical support personnel | 0.353 | 40.0 | 0.322 | 48.941 |
| Temporal | High acuity | 0.377 | 37.778 | 0.35 | 45.114 |
| | Low acuity < 60 mins | 0.168 | 86.667* | 0.126 | 69.32* |
| | Admit ED LOS < 6 hours | 0.139 | 88.889* | 0.099 | 75.033* |
| | Discharge ED LOS < 4 hours | 0.153 | 86.667* | 0.129 | 72.361* |
| | Time to triage | 0.353 | 62.222 | 0.334 | 53.917* |
| | Time to start of treatment | 0.319 | 53.333 | 0.288 | 52.762 |
| | Time to ED bed | 0.324 | 48.889 | 0.283 | 50.826 |
| | Time to treatment condition | 0.299 | 66.667* | 0.268 | 57.95* |
| Quality | Employee fatigue | 0.724 | 100.0* | 0.741 | 60.947* |
| | Employee satisfaction | 0.14 | 91.111* | 0.119 | 75.279* |
| | Medical errors | 0.597 | 64.444* | 0.602 | 52.572 |
| Outcomes | Patients hospitalized | 0.458 | 93.333* | 0.454 | 64.093* |
| | Patient transfers | 0.353 | 35.556 | 0.322 | 49.14 |
| Financial expenditures | Increase diagnostic test | 0.538 | 64.444* | 0.54 | 49.038 |
| | Increase ED treatment | 0.476 | 48.889 | 0.46 | 44.28 |
| | Increase ED revenue | 0.387 | 68.889* | 0.385 | 61.557* |
| | Increase in non-labor cost | 0.564 | 66.667* | 0.571 | 51.65 |

The values with (*) show consensus based on group opinions for each metric.

The stability results obtained with Mann-Whitney's $U$ test showed statistical significance for FDM ($z = 10.0$ and $p < 0.05$) and MOFD ($z = 62.0$ and $p < 0.05$). We did not include the Delphi method since the results do not proceed past the first round. MOFD results showed that opinion changes led to a higher level of agreement in round two.

**Table 3. Weighted sentiment scores.**

| Healthcare performance factors | Negative Score | Positive Score | Sentiment |
|---|---|---|---|
| Q18 –Capacity | 0.021 | 0.010 | Negative |
| Q24 –Temporal | 0.027 | 0.007 | Negative |
| Q26 –Quality | 0.007 | 0.010 | Positive |
| Q30 –Outcomes | 0.006 | 0.005 | Negative |
| Q32 –Financial Expenditures | 0.007 | 0.008 | Positive |

**Table 4. Second-round results for FDM & MOFD method.**

| Healthcare performance factors | Metrics | FDM | | MOFD | |
|---|---|---|---|---|---|
| | | Avg. of fuzzy numbers | Consensus (threshold >70) | Avg. of fuzzy numbers | Consensus (threshold >56) |
| Capacity | Physician staffing | 0.478 | 52.174 | 0.473 | 51.15 |
| | Midlevel provider staffing | 0.557 | 69.565 | 0.571 | 57.06* |
| | Patient acuity level | 0.643 | 86.957* | 0.691 | 56.91* |
| | Physician staffing per patient seen | 0.661 | 91.304* | 0.665 | 63.41* |
| | Nurse staffing per patient seen | -- | -- | 0.781 | 77.34* |
| | Backup physician | 0.452 | 69.565 | 0.434 | 54.08 |
| | Backup nurse | 0.576 | 60.87 | 0.585 | 49.23 |
| | Patient care compromised | -- | -- | 0.702 | 56.7* |
| | Medical support personnel | 0.557 | 52.174 | 0.567 | 45.87 |
| Temporal | High acuity <30 mins | 0.557 | 60.87 | 0.582 | 48.2 |
| | Time to start of treatment | 0.43 | 78.261* | 0.43 | 57.02* |
| | Time to ED bed | 0.443 | 56.522 | 0.44 | 43.53 |
| Quality | Medical errors | -- | -- | 0.618 | 52.25 |
| Outcomes | Patient transfers | 0.239 | 91.304* | 0.205 | 65.64* |
| Financial expenditures | Increase diagnostic test | -- | -- | 0.546 | 52.51 |
| | Increase ED treatment | 0.522 | 78.261* | 0.537 | 58.27* |
| | Increase in non-labor cost | -- | -- | 0.579 | 55.82 |

The values with (*) show consensus based on group opinions for each metric.

Furthermore, the hypothesis test demonstrated that the panelists' scores are stable, which means a convergence of the MOFD process. Finally, 21 performance indicators were retained in the FDM, and 20 were retained for MOFD to measure the performance of the EDs during a medical surge using the consensus views of the medical professionals. Fig 2 presents the final list of indicators and compares the results of the two methods.

| FDM | | In Common? | MOFD | |
|---|---|---|---|---|
| Ranking | Indicator | | Indicator | Ranking |
| 1 | Employee fatigue | TRUE | ED beds | 1 |
| 2 | ED beds | TRUE | Nurse staffing per patient seen | 2 |
| 3 | Patients hospitalized | TRUE | Employee satisfaction | 3 |
| 4 | Patient transfers | TRUE | ICU beds | 4 |
| 4 | Physician staffing per patient seen | TRUE | Admit ED LOS | 5 |
| 6 | Employee satisfaction | TRUE | Discharge ED LOS | 6 |
| 6 | ICU beds | TRUE | Low acuity | 7 |
| 8 | Admit ED LOS | TRUE | Patient transfers | 8 |
| 9 | Patient acuity level | TRUE | Patients hospitalized | 9 |
| 10 | Discharge ED LOS | TRUE | Physician staffing per patient seen | 10 |
| 10 | Low acuity | TRUE | Nurse staffing | 11 |
| 10 | Nurse staffing | TRUE | Increase ED revenue | 12 |
| 13 | Nurse staffing per patient seen | TRUE | Employee fatigue | 13 |
| 13 | Patient care compromised | TRUE | Increase ED treatment | 14 |
| 15 | Increase ED treatment | TRUE | Time to treatment condition | 15 |
| 15 | Time to start of treatment | TRUE | Midlevel provider staffing | 16 |
| 17 | Increase ED revenue | TRUE | Time to start of treatment | 17 |
| 18 | Increase in non-labor cost | FALSE | Patient acuity level | 18 |
| 18 | Time to treatment condition | TRUE | Patient care compromised | 19 |
| 20 | Increase diagnostic test | FALSE | Time to triage | 20 |
| 20 | Medical errors | FALSE | | |

**Fig 2. Ranking and comparison of indicators for the FDM and MOFD method, respectively.**

**Table 5. Comparing indicators for normal and surge conditions.**

| Rank | Normal Conditions (MOFD) | Surge Conditions (MOFD) |
|:---:|---|---|
| 1 | Increase in non-labor cost | ED beds |
| 2 | Increase in ED revenue | Nurse staffing per patient seen |
| 3 | Patients hospitalized | Employee satisfaction |
| 4 | Time to start of treatment | ICU beds |
| 5 | Medical errors | Admit ED length of stay |

Fig 2 displays the 18 out of 21 common indicators found when comparing the two methods. The MOFD method ranks ED beds, nurse staffing per patient seen, employee satisfaction, and ICU beds as the most critical indicators influencing ED performance during a surge. In contrast, FDM ranks employee fatigue, ED beds, patients hospitalized, and physician staffing per patient seen as the most critical indicators. Also, our analysis showed that certain indicators were ranked differently for the two methods. For instance, employee fatigue is a common indicator between the two methods and is ranked first in FDM and thirteen in MOFD.

ED bed is another common indicator ranked first and second in MOFD and FDM. When comparing the rankings, ED beds are essential for treating patients during a medical surge than employee fatigue based on expert opinions. Also, patients cannot get the needed treatment without beds, which strains medical personnel, affecting performance [33]. An increase in ED beds requires an increase in nurses or physicians. Still, in the current COVID-19 pandemic, this is not the case as some healthcare workers are infected, leading to a shortage of frontline medical workers who are stressed and overwhelmed [33]. As part of our survey, the medical professionals also provided their opinions on the indicators that influence the ED's performance during a normal condition. We analyzed the data using our proposed MOFD approach, and Table 5 compares the indicator identification results for normal and surge operating conditions (see S7 Table in S1 File for full results).

We observed that the top five ranked indicators based on the MOFD approach for normal and surge operating conditions are entirely different, as presented in Table 5. During normal operating conditions, an increase in non-labor costs can produce resource shortages, decreasing clinician productivity. In addition, surges create shortages in ED beds, which cause patients to wait more prolonged periods before being evaluated and treated. Depending on the given conditions in the ED, hospital administrators, healthcare quality experts, and clinicians can know what indicators to focus on to improve performance.

## Discussion

The current COVID-19 pandemic has overwhelmed various health systems around the globe, with infection and death rates increasing daily. Hospital administrators must develop daily strategies to properly manage the situation and ensure adequate allocation of the limited resources to care for patients while taking appropriate measures to ensure that the health and safety of medical professionals are not compromised. To effectively manage the performance of an ED in a surge event, it is vital to identify and select those indicators that affect its performance. However, few studies have provided a framework for selecting ED performance indicators in the event of a medical surge (i.e., epidemic or pandemic). Hence, we carried out a systematic literature review for two reasons. First, to investigate and understand the current state of practice on ED indicator identification during a medical surge. Second, to inform the selection of indicators while developing the survey. The literature review confirmed the

existence of a gap concerning evidence-based indicators needed for evaluating the performance of the ED during a surge.

In this study, we propose a consistency and consensus-based model integrated with text mining analysis to support the consensus reaching process in identifying a pertinent set of critical indicators that influence the efficacy of EDs during a medical surge. The findings indicate that the MOFD method was more reliable in analyzing the survey responses than the DM and FDM methods. The DM approach was not used past the first round as it would require multiple rounds before consensus is achieved. Multiple rounds will be time-consuming and expensive, especially when dealing with healthcare experts. The results of the DM approach confirm the drawbacks of the method's inability to handle uncertainty and linguistic information inherent in human consensus processes.

The proposed MOFD method combines the similarity and distance measures to deal with the situation when expert opinions are disjoint to achieve consistency among aggregated consensuses, thereby providing a better result for identifying ED indicators than the FDM method. Also, the inclusion of sentiment analysis in the model helps detect the polarity within the textual data and gain meaningful comprehension of participant perspectives. Therefore, the proposed methodology has the potential to identify the indicators that influence the performance of the ED during a surge while dealing with textual, qualitative, and quantitative opinions of experts.

Theoretically, these two methods have 18 indicators in common, while taking different approaches in translating experts' responses can be counted as a validation of our proposed method. Although similar in results, the main advantage of MOFD over FDM is proven to be the removal of ties and the clear rankings of indicators. The MOFD method achieved that by taking advantage of similarity coefficient and distance measure (i.e., a combination of a weighted hamming and infinum distance) to acquire a consistency index of individual panel member opinion compared to FDM that applies only the Euclidean distance measure. We discovered that ED beds, ICU beds, nurse staffing per patient seen, employee satisfaction, ED LOS, and patient acuity are essential indicators for monitoring ED performance based on stakeholder opinions during a medical surge. These indicators are priority areas for operational leadership to consider addressing during a pandemic. Implementation examples include increased access and capacity to ED beds and addressing employee safety and wellness early on.

In the current environment, multiple national health systems within the United States have recognized that employee satisfaction and fatigue have led to significant nursing shortages during the pandemic [34–36]. One potential way to rapidly increase bed capacity involves creating mobile tents/facilities or improving throughput so that admitted patients do not stay in the ED for an extended period. For a more practical validation, the performance of the FDM and MOFD methods can be evaluated under the ongoing pandemic (i.e., COVID-19). Although the study was not explicitly designed with COVID-19 in mind, it is holding up with regards to the current events as hospital administrators face challenges with performance and delivering the highest quality service to patients. To validate our findings, we look at the recent coronavirus events happening across three states in the US, namely Florida, Texas, and California. We can see that hospitals in these states are overwhelmed as the infection rates increase [37, 38]. Hospital performance is greatly affected as administrators struggle to make informed decisions on allocating ED and ICU beds to patients and assigning nurses and physicians to patients with severe illnesses, as reported by the FDM and MOFD methods. Most hospitals have exceeded capacity, cannot accept patients, and have resorted to transferring patients to other hospitals within or outside the states. A robust organizational strategy enables hospitals to prioritize and maintain critical care functions in a surge event [3].

To further validate the proposed indicators, we have applied the Bayesian change point analysis [39, 40] to a hospital management dataset provided by Henry Ford Hospital in Michigan. The time-series data contains different hospital performance measures, including admit ED LOS, number of nurses, midlevel provider staffing, number of ICU beds, time to start of treatment, and time to triage from 2019 to 2020. This dataset will further validate the significant indicators that influence ED performance before and during a medical surge. As illustrated in S8 Table in S1 File, Bayesian change point analysis is applied to investigate whether any changes occurred in the given indicators before COVID-19 (including flu season) in 2019 and during COVID-19 (including flu season) in 2020.

Changes have been detected in the nurse staffing level, mid-level providers' staffing, and the number of ICU beds used. The change in overall patient volumes and a drop in patient visits due to the COVID-19 pandemic justifies the decrease in ED patients requiring ICU beds. There is an observed change in the average time to start treatment for patients who visit the ED in November 2019, similar to March 2020. The time to treatment is based on the balance between the staffing level of nurses, physicians, and the number of patient visits per hour. Even though the number of patient visits has dropped, the staffing level has been adjusted accordingly. The only change point detected for time to triage is August 2020 due to the drop in patient levels. There is an observed change in the average ED Length of stay in August 2019 and March 2020. The 2020 change is due to an estimated 40% decline in the number of patients in the ED because of COVID-19. We can infer that the changes were mainly observed during Michigan's flu season or COVID-19 peak. These change-point analyses may shed light on the change(s) in ED performance during a medical surge event and further validate our findings from our proposed MOFD model.

Our study contributes to strategic considerations for surge events by providing a broad agreement about critical indicators that affect the operation and ability to care for patients during a medical surge. Jombart et al. [41] developed a model to forecast the critical care bed requirements for patients infected with the COVID-19 in England. Their model estimates that if the transmission of the virus increases, an increase in patients will lead to the increasing demand for more ICU beds, which puts additional pressure on bed capacity in the hospitals. This pressure affects the ED's ability to cater to the needs of the incoming patients without considering the capacity requirements of existing patients.

Another study [42] described the predicted increase in healthcare service demand related to the surge capacity of ICUs in Australia with COVID-19 admissions. Their study indicates a shortage of ICU beds, ventilators, and the need to increase the hospital workforce (i.e., registered nurses) to match the growing demand of infected patients. On the contrary [43], states that bed capacity may not be a crucial indicator during this pandemic as medical and health policy experts are more concerned with the number of ventilators. In our opinion, all these medical resources (such as beds, ventilators, drugs, personal protective equipment) are essential as they affect the ED's operations and ability to provide critical care. A shortage in any may result in poor performance, which ultimately can contribute to morbidity and mortality.

Although the set of indicators identified through the proposed approach may be perceived as of nature, the exclusion of many indicators leads to a reduced set that needs to be tracked by the hospitals for performance improvement. Our model indicators can aid ED's operations to identify bottlenecks/inefficient areas and allocate limited resources to effectively manage surges. With available data from the electronic health record system, the identified indicators can be extracted and used by health systems to assess the overall performance of EDs during a medical surge. For efficiency analysis, non-parametric methods such as data envelopment analysis can utilize our identified indicators to measure the ED's performance and identify improvement areas. The assessment results will show areas that need improvement within the

respective EDs, and strategies developed to address them while creating plans for future pandemics or disasters.

## Conclusion

In this research paper, we developed and compared three consensus-based methods (DM, FDM, and MOFD) to identify quality indicators that influence the ED's performance during a medical surge. This study contributes to the body of hospital quality research on important ED indicators, and it reaffirms the perceived importance of performance indicators such as ED beds, nurse staffing per patient seen, employee satisfaction, ICU beds in the context of a medical surge, and it sets a potential agenda for future research reporting and administrative oversight. The indicators presented in this study have face validity and can help develop and guide improvements in uniform ED data collection systems to monitor performance during a medical surge. Further rigorous assessment and evaluation of the identified indicators in hospitals are needed to improve the effectiveness, applicability, and adoption of appropriate indicators by hospital quality researchers, doctors, administrators, policymakers, and ED researchers. Most of the indicators we identified can be obtained in systems with sophisticated ED information systems. Although no set of indicators would be able to paint the full picture of a surge and its effects adequately on its own, the identified indicators can be used for ED performance assessment and to inform practices that will assist hospital administrators and health systems in being better prepared for future pandemics or disasters. In addition, other potential surge healthcare performance factors, such as patient and physician outcomes, quality, and financial measures, are studied in this work.

Although this study has successfully introduced indicators that can improve the performance of EDs during a medical surge, we note the following limitations. First, this study is done with health professionals based in Michigan. This sample limitation may affect the indicator selection, limiting us from generalizing the results to a national or international level. For future work, larger sample size will be constructed by inviting national and international medical professionals to participate in the study. Second, we observed missing values in some aspects of the participant responses. Although missing values may reduce the statistical power and the representativeness of the sample, we solve this problem by imputing a neutral response to the missing data point. Third, a relatively low response rate of 51.11% from the second-round survey participants may lead to non-response bias, affecting how well the data represents the survey population. Fourth, the study may be limited as we covered only the relevant set of indicators. Other indicators such as early mortality after arrival, number of tests performed, number of personal protective equipment, and number of ventilators represent essential indicators to measure ED performance. Due to the lack of data and time sensitivity of this research, the authors focused on performance improvement-related indicators. Additionally, as part of our future work, we will investigate the use of Soft Operations Research or Problem Structuring methods such as Soft Systems Methodology, the Strategic Choice Approach, and Strategic Options Development and Analysis to discuss the meaning of each indicator that influence ED performance during surge conditions with medical professionals. Next, we will perform an in-depth literature review on the best multicriteria clustering approach for the work.

## Supporting information

**S1 File. Detailed methods, results, and tables.**
(DOCX)

## Author Contributions

**Conceptualization:** Egbe-Etu Etu, Leslie Monplaisir, Celestine Aguwa, Joseph Miller.

**Data curation:** Egbe-Etu Etu, Suzan Arslanturk, Ihor Markevych, Joseph Miller.

**Formal analysis:** Egbe-Etu Etu, Suzan Arslanturk, Sara Masoud, Ihor Markevych.

**Funding acquisition:** Egbe-Etu Etu, Leslie Monplaisir, Celestine Aguwa, Joseph Miller.

**Investigation:** Egbe-Etu Etu, Celestine Aguwa, Joseph Miller.

**Methodology:** Egbe-Etu Etu, Leslie Monplaisir, Celestine Aguwa, Suzan Arslanturk, Sara Masoud, Ihor Markevych.

**Project administration:** Joseph Miller.

**Resources:** Joseph Miller.

**Software:** Egbe-Etu Etu, Suzan Arslanturk, Sara Masoud, Ihor Markevych.

**Supervision:** Celestine Aguwa, Sara Masoud, Joseph Miller.

**Validation:** Suzan Arslanturk, Sara Masoud.

**Visualization:** Suzan Arslanturk, Sara Masoud.

**Writing – original draft:** Egbe-Etu Etu, Suzan Arslanturk, Ihor Markevych.

**Writing – review & editing:** Leslie Monplaisir, Celestine Aguwa, Suzan Arslanturk, Sara Masoud, Joseph Miller.

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
