## [Decision Letter · Decision Letter 0]

12 Jan 2022

PONE-D-21-39883Identifying Indicators Influencing Emergency Department Performance during a Medical Surge: A Consensus-Based Modified Fuzzy Delphi ApproachPLOS ONE

Dear Dr. Etu,

Thank you for submitting your manuscript to PLOS ONE. After careful consideration, we feel that it has merit but does not fully meet PLOS ONE’s publication criteria as it currently stands. Therefore, we invite you to submit a revised version of the manuscript that addresses the points raised during the review process. Please submit your revised manuscript by Feb 26 2022 11:59PM. If you will need more time than this to complete your revisions, please reply to this message or contact the journal office at plosone@plos.org. Please include the following items when submitting your revised manuscript:A rebuttal letter that responds to each point raised by the academic editor and reviewer(s). You should upload this letter as a separate file labeled 'Response to Reviewers'.A marked-up copy of your manuscript that highlights changes made to the original version. You should upload this as a separate file labeled 'Revised Manuscript with Track Changes'.An unmarked version of your revised paper without tracked changes. You should upload this as a separate file labeled 'Manuscript'.

We look forward to receiving your revised manuscript.

Kind regards,

Yong-Hong Kuo

Academic Editor

PLOS ONE

Journal Requirements:

Additional Editor Comments:

The manuscript has been reviewed by two experts in the area. Both of the referees appreciate the value of the paper. However, Reviewer 2 has some concerns about the organization of the manuscript and the writing. I have gone through the manuscript and agree with the referees. This study on ED performance is important and has value. I also agree with Reviewer 2 that the organization of the work shall be strengthened to reflect the contributions and key messages of the work (report of results / development of the methodology).

In addition, I have one minor suggestion. The majority of the references used in this study are not recent. To review the state of the art and show the timeliness of the study, the authors may include some more recent relevant studies to interest the reader. I realize some of the recent related studies, e.g.,:

Ansah, J. P., Ahmad, S., Lee, L. H., Shen, Y., Ong, M. E. H., Matchar, D. B., & Schoenenberger, L. (2021). Modeling Emergency Department crowding: Restoring the balance between demand for and supply of emergency medicine. Plos one, 16(1), e0244097.

Barak-Corren, Y., Chaudhari, P., Perniciaro, J., Waltzman, M., Fine, A. M., & Reis, B. Y. (2021). Prediction across healthcare settings: a case study in predicting emergency department disposition. NPJ digital medicine, 4(1), 1-7.

Benevento, E., Aloini, D., & Squicciarini, N. (2021). Towards a real-time prediction of waiting times in emergency departments: A comparative analysis of machine learning techniques. International Journal of Forecasting.

Kuo, Y. H., Chan, N. B., Leung, J. M., Meng, H., So, A. M. C., Tsoi, K. K., & Graham, C. A. (2020). An integrated approach of machine learning and systems thinking for waiting time prediction in an emergency department. International journal of medical informatics, 139, 104143.

Kuo, Y. H., Rado, O., Lupia, B., Leung, J. M., & Graham, C. A. (2016). Improving the efficiency of a hospital emergency department: a simulation study with indirectly imputed service-time distributions. Flexible Services and Manufacturing Journal, 28(1-2), 120-147.

Tang, K. J. W., Ang, C. K. E., Constantinides, T., Rajinikanth, V., Acharya, U. R., & Cheong, K. H. (2021). Artificial intelligence and machine learning in emergency medicine. Biocybernetics and Biomedical Engineering, 41(1), 156-172.

Vanbrabant, L., Braekers, K., Ramaekers, K., & Van Nieuwenhuyse, I. (2019). Simulation of emergency department operations: A comprehensive review of KPIs and operational improvements. Computers & Industrial Engineering, 131, 356-381.

Reviewers' comments:

Reviewer's Responses to Questions

**Comments to the Author**

1. Is the manuscript technically sound, and do the data support the conclusions?

Reviewer #1: Yes

Reviewer #2: Partly

2. Has the statistical analysis been performed appropriately and rigorously? 

Reviewer #1: I Don't Know

Reviewer #2: Yes

3. Have the authors made all data underlying the findings in their manuscript fully available?

Reviewer #1: No

Reviewer #2: Yes

4. Is the manuscript presented in an intelligible fashion and written in standard English?

Reviewer #1: Yes

Reviewer #2: No

5. Review Comments to the Author

Reviewer #1: Thanks for the opportunity. This was a well-written and easy-to-follow article with a significant scientific value.

Numerous methods have been described in the literature to identify Emergency Department performance throughout years. These methods are useful as a guide to enhance quality of care in workplace productivity in Emergency departments. This is a well-written and well-designed article on indicators of Emergency department performance. Particularly in the COVID-19 era, we hope that number of such researches increase.

Best regards.

Reviewer #2: The overall message of this project is valuable (identifying perfomance indicators for ED's during a surge) and is highly relevant given the pandemic, however it requires massive editing. As written, it can be difficult to tell if it is a traditional manuscript reporting results or a methodology paper. Some of this has to do with verb tense agreement (recommend using simple past tense throughout). It also requres an overall edit for grammar, targeting both verb tense correction and elimination of sentence fragments.

I would find it acceptable if the authors described the problem and said that they used a fuzzy Delphi approach as their methodology. Pages 4-8 can be deleted or significantly shortened. Similarly, pages 9-12 can be significantly condensed to only give key details on how the process was implemented. Pages 13-22 provide more substantial content (results) but could still be edited to minimize overly detailed references to methods. A major overhaul in these sections could significantly improve the clarity and readability of the manuscript.

6. PLOS authors have the option to publish the peer review history of their article (what does this mean?). If published, this will include your full peer review and any attached files.

Reviewer #1: No

Reviewer #2: No

---

## [Author Response · Author response to Decision Letter 0]

1 Feb 2022

Dear editor and reviewers,

Thank you for giving us the opportunity to submit a revised draft of our manuscript titled “Identifying Indicators Influencing Emergency Department Performance during a Medical Surge: A Consensus-Based Modified Fuzzy Delphi Approach”. We appreciate the time and effort that you and the reviewers have dedicated to providing your valuable feedback on our manuscript. We are grateful to the reviewers for their insightful comments. We have been able to incorporate changes to reflect all of the suggestions. Also, we have highlighted the changes within the manuscript.

---

## [Decision Letter · Decision Letter 1]

23 Feb 2022

Identifying Indicators Influencing Emergency Department Performance during a Medical Surge: A Consensus-Based Modified Fuzzy Delphi Approach

PONE-D-21-39883R1

Dear Dr. Etu,

We’re pleased to inform you that your manuscript has been judged scientifically suitable for publication and will be formally accepted for publication once it meets all outstanding technical requirements.

Kind regards,

Yong-Hong Kuo

Academic Editor

PLOS ONE

Additional Editor Comments (optional):

Based on the referees' recommendations, I recommend Accept.

Reviewers' comments:

Reviewer's Responses to Questions

**Comments to the Author**

1. If the authors have adequately addressed your comments raised in a previous round of review and you feel that this manuscript is now acceptable for publication, you may indicate that here to bypass the “Comments to the Author” section, enter your conflict of interest statement in the “Confidential to Editor” section, and submit your "Accept" recommendation.

Reviewer #1: All comments have been addressed

Reviewer #2: All comments have been addressed

2. Is the manuscript technically sound, and do the data support the conclusions?

Reviewer #1: Yes

Reviewer #2: Yes

3. Has the statistical analysis been performed appropriately and rigorously? 

Reviewer #1: Yes

Reviewer #2: Yes

4. Have the authors made all data underlying the findings in their manuscript fully available?

Reviewer #1: Yes

Reviewer #2: Yes

5. Is the manuscript presented in an intelligible fashion and written in standard English?

Reviewer #1: Yes

Reviewer #2: Yes

6. Review Comments to the Author

Reviewer #1: (No Response)

Reviewer #2: (No Response)

7. PLOS authors have the option to publish the peer review history of their article (what does this mean?). If published, this will include your full peer review and any attached files.

Reviewer #1: No

Reviewer #2: No

---

## [Editor Report · Acceptance letter]

28 Feb 2022

PONE-D-21-39883R1 

Identifying Indicators Influencing Emergency Department Performance during a Medical Surge: A Consensus-Based Modified Fuzzy Delphi Approach 

Dear Dr. Etu:

I'm pleased to inform you that your manuscript has been deemed suitable for publication in PLOS ONE. Congratulations! Your manuscript is now with our production department. 

Kind regards, 

on behalf of

Dr. Yong-Hong Kuo 

Academic Editor

PLOS ONE